# Gas Separation Silica Membranes Prepared by Chemical Vapor Deposition of Methyl-Substituted Silanes

**DOI:** 10.3390/membranes9110144

**Published:** 2019-11-03

**Authors:** Harumi Kato, Sean-Thomas B. Lundin, So-Jin Ahn, Atsushi Takagaki, Ryuji Kikuchi, S. Ted Oyama

**Affiliations:** 1Department of Chemical System Engineering, The University of Tokyo, 7-3-1 Hongo, Bunkyo-ku, Tokyo 113-8556, Japan; phthalmon@yahoo.co.jp (H.K.); lundin.sean@gmail.com (S.-T.B.L.); ssorobong@gmail.com (S.-J.A.); atakagak@cstf.kyushu-u.ac.jp (A.T.); rkikuchi@chemsys.t.u-tokyo.ac.jp (R.K.); 2Department of Chemical Engineering, Virginia Tech, Blacksburg, VA 24061, USA; 3College of Chemical Engineering, Fuzhou University, Fuzhou 350116, China

**Keywords:** silica-based membrane, hydrogen separation, CVD, pore size control, tetramethyl orthosilicate, methyltrimethoxysilane, dimethyldimethoxysilane, separation mechanism

## Abstract

The effect on the gas permeance properties and structural morphology of the presence of methyl functional groups in a silica membrane was studied. Membranes were synthesized via chemical vapor deposition (CVD) at 650 °C and atmospheric pressure using three silicon compounds with differing numbers of methyl- and methoxy-functional groups: tetramethyl orthosilicate (TMOS), methyltrimethoxysilane (MTMOS), and dimethyldimethoxysilane (DMDMOS). The residence time of the silica precursors in the CVD process was adjusted for each precursor and optimized in terms of gas permeance and ideal gas selectivity criteria. Final H_2_ permeances at 600 °C for the TMOS-, MTMOS-, and DMDMOS-derived membranes were respectively 1.7 × 10^−7^, 2.4 × 10^−7^, and 4.4 × 10^−8^ mol∙m^−2^∙s^−1^∙Pa^−1^ and H_2_/N_2_ selectivities were 990, 740, and 410. The presence of methyl groups in the membranes fabricated with the MTMOS and DMDMOS precursors was confirmed via Fourier-transform infrared (FTIR) spectroscopy. From FTIR analysis, an increasing methyl signal in the silica structure was correlated with both an improvement in the hydrothermal stability and an increase in the apparent activation energy for hydrogen permeation. In addition, the permeation mechanism for several gas species (He, H_2_, Ne, CO_2_, N_2_, and CH_4_) was determined by fitting the gas permeance temperature dependence to one of three models: solid state, gas-translational, or surface diffusion.

## 1. Introduction

Silica membranes are an attractive emerging gas separation technology because of advantages over palladium membranes such as chemical resistance, stability, and low cost [1,2,3,4]. Additionally, silica membranes exhibit high perm-selectivity at high temperature and a tunable morphology that can be modified for a variety of gas separations such as CO_2_/CH_4_ [5,6,7]. Silica and silica-containing membranes were studied for use in membrane reactors [8,9,10] and are beginning to be applied industrially in gas separation systems [11]. A drawback of silica membranes is their poor hydrothermal stability [12], which is detrimental in industrial applications with steam, such as pervaporation of aqueous mixtures and hydrogen production by steam reforming.

To improve the hydrothermal stability of silica membranes, various silicon compounds with different functional groups were studied such as silanes containing organic functional groups (methyl [13,14,15,16], propyl [15], hexyl [17], phenyl [15,18,19,20], and vinyl [21,22]) attached directly to a single silicon atom, and alkoxy siloxanes bridged with ≡Si–R–Si≡ structures (e.g., bis(triethoxysilyl)ethane) [23,24,25,26]. However, there were few reports focused on the number of the functional groups in a silicon compound.

One functional group that was studied is the addition of phenyl groups, such as dimethoxydiphenylsilane [27,28,29,30]. Ohta et al. [19] and Zhang et al. [31,32], for example, developed silica membranes deposited with silicon compounds containing zero, one, two, or three phenyl groups. Notably, the increasing number of phenyl groups was correlated with an increase in the estimated pore size in the membrane, increasing from ca. 0.3 nm for the tetraethyl orthosilicate-derived membrane to ca. 0.49 nm for the membrane derived from methoxy(triphenyl)silane. This increase in pore size has the potential to increase H_2_ permeance while maintaining high selectivity toward larger organic species such as methylcyclohexane (kinetic diameter: 0.60 nm) and toluene (0.59 nm).

Nomura et al. [15,33] studied methyl groups in silicon compounds and developed silica membranes deposited with five kinds of silica precursors including tetramethyl orthosilicate (TMOS), methyltrimethoxysilane (MTMOS), dimethyldimethoxysilane (DMDMOS), and trimethylmethoxy-silane (TMMOS) using the chemical vapor deposition (CVD) method, in which oxygen was added to the synthesis mixture. In that work, with increasing number of methyl groups in the silica precursor, the H_2_ permeance reached a maximum of 9.0 × 10^−7^ mol∙m^−2^∙s^−1^∙Pa^−1^ at 600 °C using DMDMOS, but then declined for TMMOS. The activation energies calculated from Arrhenius plots were 10 kJ∙mol^−1^ for TMOS, 9.3 kJ∙mol^−1^ for MTMOS, 7.9 kJ∙mol^−1^ for DMDMOS, and 3.9 kJ∙mol^−1^ for TMMOS. The authors claimed that the deposition mechanism of TMMOS appeared to be different from that of TMOS or DMDMOS. Finally, they concluded that the TMOS-derived membrane was the best among the five membranes because of the smooth membrane surface, the dense silica, and the lowest activation energy for H_2_ permeation.

The main objective of this study is to determine the effect of the number of methyl groups in a silica precursor on the synthesized membrane properties. The work here builds upon the previous research of Nomura et al. by conducting CVD under different conditions than those used previously, and uses additional characterization techniques, such as Fourier-transform infrared spectroscopy (FTIR), to probe deeper into the microstructure of these silica membranes. In this study, membrane features such as the surface morphology, the presence of surface functional groups, and the silica matrix structure were studied. Furthermore, the gas permeation properties were investigated for various gas species not tested by Nomura et al., and all results were fitted to known gas diffusion mechanisms in an effort to gain insight into the mechanism of transport across a wide array of gas species. Before comparison between the silica precursors, adjustments in the deposition conditions were carried out in order to develop defect-free membranes.

## 2. Materials and Methods

### 2.1. Membrane Synthesis

Figure 1a shows a schematic structure of the prepared silica membranes, which consisted of a commercial α-alumina support, a γ-alumina intermediate layer, and a topmost silica layer. The γ-alumina intermediate layer was applied via boehmite sols using the sol–gel technique to form a graded structure, as described in the work of Gu and Oyama [34]. The boehmite sols were synthesized through the hydrolysis of aluminum isopropoxide and peptization by nitric acid. This involved mixing 0.3 mol of aluminum isopropoxide (Sigma-Aldrich, St. Louis, MI, USA, >98%) into 50 mL of distilled water and stirring for 24 h at 98 °C. Then, nitric acid was slowly added (80 nm; H^+^/Al = 0.025, 40 nm; H^+^/Al = 0.070) and mixed for a further 24 h at 98 °C to induce peptization (oligomerization). Afterward, a solution of 0.7 g of polyvinyl alcohol (Polyscience, Warrington, PA, USA, M.W. = 78,000) in 20 mL of distilled water was added to control viscosity and keep the boehmite colloidal sols stable. Finally, water was added to adjust the total volume to 200 mL, and the mixture was stirred for 3 h at 70 °C. The particle size distributions were determined by a dynamic light scattering analyzer (Horiba LB-550, Japan) and confirmed to have averages of 40 nm and 80 nm.

Before deposition of the selective silica layer, the support was sealed into solid alumina tubes, and the γ-alumina intermediate layer was deposited as depicted in Figure 1b. Firstly, two non-porous alumina tubes (Sakaguchi E.H Voc Co., Tokyo, Japan, I.D. = 4 nm, O.D. = 6 mm, length = 200 mm) were connected on each end of an α-alumina asymmetric porous support (Noritake Co., Nagoya, Japan, I.D. = 4 mm, O.D. = 6 mm, length = 30 mm, nominal pore size = 60 nm) with a glass paste (Nippon Electric Glass Co., Ltd., Otsu, Japan) and fired at 1000 °C for 10 min with a heating and cooling rate of 5 °C∙min^−1^. Then, the prepared tube was dipped into the 80-nm boehmite sol for 10 s with the outside surface wrapped in polytetrafluoroethylene (PTFE) tape to avoid contacting the sol suspension. Afterward, the tube was dried in a cleanroom for 4 h at room temperature and calcined at 650 °C for 3 h using a heating and cooling rate of 1.5 °C∙min^−1^. Finally, this dip-coating/calcining cycle was repeated with the 40-nm sol.

After preparation of the γ-alumina intermediate layer, a microporous silica layer was deposited by a one-sided diffusion chemical vapor deposition (CVD) method as depicted in Figure 1c. In this study, tetramethyl orthosilicate (TMOS, TCI, Tokyo, Japan, >99%), methyltrimethoxysilane (MTMOS, TCI, Tokyo, Japan, >98%), and dimethyldimethoxysilane (DMDMOS, TCI, Tokyo, Japan, >98%) were used as the silica precursors. Table 1 summarizes the structure and vapor pressure of the three precursors; note that the vapor pressure increases with methyl substitution. The precursor temperature in the bubbler was maintained at 25 °C and argon (>99.998%) was used as the carrier gas.

The membrane support was fixed coaxially inside a stainless-steel reactor by using machined Swagelok fittings and polytetrafluoroethylene (PTFE) ferrules before being heated in an electric furnace at 1.5 °C∙min^−1^ to 650 °C. Gas lines from the bubbler to the reactor were also heated by a ribbon heater (ca. 120 °C) to prevent condensation of the precursor compounds. Argon was flown through the heated bubbler as the carrier gas and was mixed with a secondary argon dilution stream before being fed to the interior of the membrane support. The concentrations of silanes in the gas feed were 0.71 mol.% for TMOS, 1.6 mol.% for MTMOS, and 3.7 mol.% for DMDMOS. The total flow rate, or residence time (RT), of the CVD gas mixture was tuned for each deposition, as discussed in Section 3.1. A balance argon gas stream of the same flow rate as the inner mixtures was introduced to the outer shell of the reactor in order to maintain pressure balance between the inside and outside of the support tube. Due to the different rates of decomposition for each precursor and residence time, the CVD time was altered for each membrane synthesis.

### 2.2. Membrane Permeance Tests

The flow rates of the permeated gases were measured either by a digital flow meter (GF1010, GL Science) for flow rates above 1 mL∙min^−1^, or micro gas chromatography (Micro GC, TCD, Agilent 490, GL Science—Molecular sieve 5A column for N_2_ and Porapak Q column for CO_2_, CH_4_, and C_2_H_6_) for flow rates below 1 mL∙min^−1^. GC measurements were conducted using a 50-mL∙min^−1^ argon sweep, and individual gas flow rates were converted using the signal intensity of calibrated peak areas. For flow meter measurements, each permeance was allowed to stabilize for approximately 5 min, and the error in each measurement was estimated to be less than 2%. GC measurements were conducted at least three times for each gas permeance, and the typical error was less than 5%. Single gas permeance, ${\bar{P}}_{i}$, was calculated as
(1)$${\bar{P}}_{i}\;=\;\frac{F_{i}}{A\Delta p_{i}},$$ where $F_{i}$ is the molar flow rate of component *i*, A is the effective membrane area, and Δ*p_i_* is the trans-membrane partial pressure difference. The effective membrane area, *A*, was calculated using
(2)A =πL(r1−r2)ln(r1r2), where *L* is the length of the membrane, *r*_1_ is the outer diameter, and *r*_2_ is the inner diameter.

Because the silica membranes were supported on an alumina substrate and intermediate layer, the resistance to flow through the support needed to be subtracted. The assumption was that each layer acts independently and, thus, does not have any mass transfer process between layers. This assumption comes from the notion that transport through the alumina intermediate layer is simply Knudsen diffusion with no particular surface interactions. The subtraction was done by using the inverse of the permeance of the membrane with the γ-alumina intermediate layer but before CVD treatment, ${\bar{P}}_{Before\;CVD}^{-1}$, as the permeation resistance of the support. Then, the permeance only through the silica layer, ${\bar{P}}_{Silica\;layer}$, was modified as explained in Oyama et al. [2] according to
(3)$$\frac{1}{{\bar{P}}_{Silica\;layer}}=\;\frac{1}{{\bar{P}}_{After\;CVD}}\;-\;\frac{1}{{\bar{P}}_{Before\;CVD}}.$$

Pure gas permeance measurements were made for several gases across a wide range of kinetic diameters: He (0.255 nm), Ne (0.275 nm), H_2_ (0.289 nm), CO_2_ (0.33 nm), N_2_ (0.364 nm), CH_4_ (0.38 nm), and C_2_H_6_ (0.39 nm). All gases were above 99.99% purity and were used at all temperatures tested except ethane, which was measured only at the lowest temperature (300 °C) to avoid thermal decomposition. To avoid any potential effects from membrane thermal instability, the gas permeance was measured according to the following temperature order: 650 °C, 500 °C, 300 °C, 400 °C, 600 °C, and 650 °C. The obtention of smooth curves was evidence that the membrane was stable during the course of the measurements.

Hydrothermal stability is a known issue in silica membranes because the presence of water vapor catalyzes the formation and condensation of silanol groups and causes a densification of the silica structure [35]. To test for variances in hydrothermal stability between silica precursors, the stability under hydrothermal conditions was checked at 650 °C by flowing 10 mL∙min^−1^ of argon gas through a bubbler with water at 56 °C to produce 16 mol.% (6.6 μmol∙s^−1^) water vapor for 96 h. Simultaneously, 15 mL∙min^−1^ of argon was introduced on the opposite side of the membrane as a balance gas.

### 2.3. Membrane Characterization

Membrane surface and cross-sectional analysis was performed using a field-emission scanning electron microscope (FE-SEM Hitachi S-900). After testing, the membranes were prepared for SEM by fracturing and cutting the support with a diamond saw. The samples were then coated with Pt by ion sputtering (Hitachi E-1030) using a 15-mA current for 15 s to prevent surface charging and image drift.

Fourier-transform infrared spectroscopy (FTIR) was performed on the membranes ex situ after permeance testing using a JASCO FT/IR-6100 spectrometer equipped with an MCT detector. Samples were prepared by scraping the membrane from the surface of the α-alumina support and mixing with potassium bromide as an inert filler. The mixed powder was then pelletized into a 1-cm diameter disc at 40 MPa and placed in the open-air chamber of the FTIR. Spectra were recorded in absorbance mode at room temperature with a resolution of 4 cm^−1^ using 200 scans from 4000 cm^−1^ to 700 cm^−1^.

## 3. Results and Discussion

### 3.1. Effect of Precursor Residence Time in CVD

In CVD systems, the residence time (RT) of the precursor can have a significant effect on the final deposited structure. For this reason, the residence time was varied for each of the precursors to optimize both the permeance and perm-selectivity of the resultant membranes. The residence time is inversely proportional to flow rate and is defined as
(4)$$RT\left[s\right]=\;\frac{A\left[{\mathrm{cm}}^{2}\right]\times L\left[\mathrm{cm}\right]}{Q\left[\mathrm{mL}\cdot s^{-1}\right]},$$ where *A* is the inner cross-sectional area of the support, *L* is the membrane length, and *Q* is the volumetric flow rate of the total gas mixture being fed.

Here, the MTMOS membrane fabrication is shown as an example, and Figure 2 shows the changes in H_2_ and N_2_ permeances during MTMOS deposition for three different RTs (0.4 s, 0.8 s, and 1.3 s). Although the final N_2_ permeance did not vary significantly, the required CVD time was significantly different for each retention time and there was no trend in the results, and this can be explained by a tradeoff between different factors. These factors are residence time (flow rate) and total reaction time. Normally, short residence times require longer CVD times, because the high flows carry away the reactant and more time is needed to plug the pores. Conversely, high residence times require shorter CVD times as there is more opportunity for reaction. However, with short residence times and high flows, the diffusion boundary layers are smaller and the reactant is quickly replenished. This enhances mass transfer and allows for shorter CVD times to suffice. Of course, the opposite holds for high residence times, meaning there is a tradeoff between macroscopic reactant supply and microscopic mass transfer.

The residence time affects the morphology of the membranes through the mechanism of growth. Examination of the surface by scanning electron microscopy (SEM) showed that short residence times produced larger grains (Figure 3, top) and longer residence times produced smaller grains (Figure 3, middle and bottom). This is because the short residence time is associated with a longer reaction time, and this allows for growth of grains and the formation of a smooth surface. With longer residence times, the opposite holds because reaction times are shorter which prevents the growth of grains and results in the formation of rougher surfaces.

Unfortunately, the cross-sectional images could not be magnified enough to show the thickness of the membranes precisely. From the images available, the thickness appeared to be about 50 nm, but it was not possible to determine precise thicknesses at the resolution of these images.

Figure 4a summarizes the permeance of several gases at 300 °C before and after CVD, and Table 2 summarizes the final membrane properties. For the alumina layer before CVD, the permeances of all the gases were very high (10^−5^ to 10^−6^ mol∙m^−2^∙s^−1^∙Pa^−1^). As shown in Figure 4b, these followed a Knudsen mechanism as shown by the dependence of permeance on the inverse square root of mass. For the membranes prepared by CVD, note that the selectivity for the smaller molecules (e.g., N_2_/C_2_H_6_) was the lowest for the RT = 1.3 s membrane. Because of the roughness of the surface of the RT = 1.3 s membrane, a few large defects remained after CVD that allowed C_2_H_6_ molecules to permeate, resulting in the small permeance difference between N_2_ (kinetic diameter of 0.364 nm) and C_2_H_6_ (0.39 nm). From this analysis, the RT = 0.8 s condition was found to be most advantageous, giving the highest final selectivity. Interestingly, this was also the membrane with the slowest reaction time (Figure 2), but intermediate sized grains (Figure 3, middle). This suggests that the tradeoff between macroscopic reactant supply and microscopic mass transfer explained before led to a non-linear grain growth rate. When considering the differences between RT = 0.4 s and RT = 0.8 s in H_2_ permeance (1.12 × 10^−8^ vs. 1.14 × 10^−8^ mol∙m^−2^∙s^−1^∙Pa^−1^) and H_2_/N_2_ selectivity (210 vs. 400), the significant difference was the greater reduction in N_2_ permeance for the RT = 0.8 s condition. Overall, however, there appeared to be only a slight variance in final properties as compared to the significant change from the RT = 1.3 s condition. This suggests that, as long as the growth rate is slow, good results can be obtained, but the RT = 1.3 s caused such a rapid growth rate that non-optimal results were obtained. From these results, the RT = 0.8 s result was considered optimal not just because of the higher H_2_/N_2_ selectivity, but because it used less silica precursor than the RT = 0.4 s condition. Although not shown here, a similar trend was observed for both the DMDMOS- and TMOS-derived membranes; the H_2_ permeance remained nearly identical, and the primary difference in membrane performance was due to the reduction of permeance for the larger gases such as N_2_. From these results, the best residence times were determined to be 0.6 s for DMDMOS and 0.8 s for TMOS.

### 3.2. Effect of Methyl-Substituted Methoxysilanes

The CVD results for the TMOS-, MTMOS- and DMDMOS-derived membranes are shown in Figure 5 and Figure 6. While the TMOS-derived membrane formed quickly, the addition of the methyl groups on MTMOS and DMDMOS significantly increased the required CVD time to sufficiently close the pores and inhibit the flow of the larger molecules (e.g., N_2_).

Additionally, SEM surface images (Figure 7) show that the particles on the surface became smaller with increasing number of methyl groups in the silica precursor. This change in particle size also correlated with a decrease in H_2_/N_2_ selectivity (Table 3) and was consistent with the residence time analysis from the previous section. A notable difference with the work of Nomura et al. [33] was that the worst performing membrane from this study, the DMDMOS-derived membrane, was actually the best performing membrane from the previous study. This may suggest that differences in the synthesis conditions can have significant effects on the final membrane performance properties. Two notable differences existed in the CVD conditions between this study and that of Nomura et al. Firstly, whereas this study conducted CVD at 650 °C, Nomura et al. used a 500 °C deposition temperature because they found that the silica layer cracked when deposited at 600 °C. This was because they used O_2_ during the CVD process, which was the second major difference in CVD conditions as compared to the current study, and they suggested that the fast reaction rate caused the substrate to break due to the heat of reaction.

The important point is the two key differences in this study compared to Nomura et al. were a lower CVD temperature and the use of O_2_ as an oxidant to decompose the DMDMOS precursor. This could result in two major differences: (1) lower CVD temperatures are well known to create looser silica networks that tend to increase permeances, which can explain the higher H_2_ permeance compared to the current study, and (2) the use of O_2_ suggests the methyl groups attached to DMDMOS would not likely remain within the silica network, which may also increase the H_2_ permeance compared to the current study. As shown in Section 3.3, this study confirmed the presence of methyl groups in the final structures, but no such evidence was presented in the studies conducted by Nomura et al.

### 3.3. FTIR Spectroscopy

FTIR measurements were conducted for the three membranes to confirm the presence of methyl groups in the final membrane structure (Figure 8). The large and broad peak around 3400 cm^−1^ indicates that a significant number of –OH groups existed in each of the membranes. However, a significant difference was observed in the region around 2900 cm^−1^, which became noticeable when the TMOS-derived membrane signal was used as the background. Figure 8b shows the signal intensity of the MTMOS- and DMDMOS-derived membranes after subtracting the spectra of the TMOS-derived membrane. This allowed a feature to become visible at ca. 2900 cm^−1^ for both the MTMOS- and DMDMOS-derived membrane signals. Vibrations in this region are characteristic of C–H bonds, which indicate the presence of methyl groups. Notably, the use of the TMOS-derived membrane as the background meant that the presence of these features could not be explained by low levels of adsorbed methane on the silica surface. Thus, this suggests that the methyl groups in the MTMOS and DMDMOS precursors remained after deposition. Furthermore, the peak signal was higher for the DMDMOS-derived membrane, which showed that higher numbers of methyl groups in the precursor also led to more C–H bonds being present after membrane formation via CVD.

### 3.4. Hydrothermal Stability Test

The hydrothermal stability was tested for the three membranes, and the results are shown in Figure 9. Under hydrothermal conditions, the H_2_ and N_2_ permeance decreased during the 96-h exposure, with the largest permeance changes occurring in the first 24 h. After that, the H_2_ permeance continued to decrease gradually, while the N_2_ permeances appeared stable (Table 4). Importantly, the decrease in H_2_ permeance was inversely proportional to the presence of methyl groups; the decrease was highest for the TMOS-derived membrane and lowest for the DMDMOS-derived membrane.

Unlike the H_2_ permeance changes, the N_2_ permeance changes did not follow a clear trend. It was previously reported that exposure to water vapor causes sintering of the γ-alumina intermediate layer, which can result in a widening of defects and an increase in N_2_ permeance [12]. This suggests that two opposing mechanisms can occur when a silica membrane is exposed to water vapor: the permeance can decline due to densification of the silica layer, or the permeance can increase due to changes in the γ-alumina intermediate layer. It is likely that both effects occurred in this study, but to differing degrees on each membrane. Furthermore, the silica layers in these membranes may have been dense enough to not allow significant water vapor to reach the intermediate layer. This may explain why the N_2_ permeance decreased for each membrane, but also why there was no clear trend with the presence of methyl groups in the structure.

### 3.5. Gas Diffusion Mechanism Analysis

Several gas diffusional mechanisms are possible in microporous networks, including Knudsen diffusion, surface diffusion, gas-translational diffusion, and solid-state diffusion. Knudsen diffusion occurs when the mean free path is longer than the diameter of the pores [36], and the gas permeance is defined by
(5)$${\bar{P}}_{K}\;=\;\frac{\varepsilon d_{P}}{\tau L}{\left(\frac{8}{9\pi MRT}\right)}^{1/2}$$ where *d_P_* (m) is the pore diameter, *L* (m) is the membrane thickness, *M* (kg∙mol^−1^) is the permeate molecular weight, *R* (J∙mol^−1^∙K^−1^) is the gas constant, and *T* (K) is the absolute temperature.

Surface diffusion can add to the flow through pores if interactions between the gas molecules and the wall are strong enough to prevent the molecules from desorbing into the gaseous phase [37,38]. The adsorbed molecules jump to other adsorption sites if they have a higher energy than the adsorption energy [39], giving a permeance described by
(6)$${\bar{P}}_{\mathrm{SD}}\;=\;P_{0}\exp\left(\frac{-\Delta H_{a}-\Delta E_{\mathrm{SD}}}{RT}\right)$$ where $\Delta H_{a}$ (J mol^−1^) is the heat of adsorption, and $\Delta E_{\mathrm{SD}}$ (J mol^−1^) is the energy barrier to jump to other adsorption sites.

In microporous silica membranes, gas-translational diffusion becomes very important due to the small size of the pore network. When the pore network is of the same size as the diffusing gas molecules, the diffusing gas molecules that have enough energy to escape the surface potential cannot readily do so due to the presence of a pore wall on the other side [2]. The permeance calculation in this case becomes a combination of the Knudsen diffusion and surface diffusion models and is called an activated Knudsen diffusion model or gas-translational model described by
(7)P¯GT = εdpρgτL(8πMRT)12exp(−ΔERT) where the contributions of both Knudsen and surface diffusion are apparent.

Lastly, solid-state diffusion can add to the permeance of gases in glass networks due to the existence of solubility sites in the silica. The behavior of the gas molecules is similar to that of surface diffusion, except that, instead of small pores, solubility sites are assumed to exist in the solid-state diffusion model [40,41,42]. The permeance equation is
(8)$${\bar{P}}_{\mathrm{SS}}\;=\;\frac{d^{2}h^{2}}{6L}{\left(\frac{1}{2\pi mkT}\right)}^{\frac{3}{2}}\left(\frac{\sigma h^{2}}{8\pi^{2}IkT}\right)\frac{N_{S}}{N_{A}}\frac{1}{{\left(e^{h\nu^{*}/2kT}-e^{-h\nu^{*}/2kT}\right)}^{2}}e^{-\frac{\Delta E_{\mathrm{SS}}}{RT}}$$ where *d* (m) is the jumping distance between sorption sites in the structure, *h* (m^2^∙kg∙s^−1^) is Planck’s constant, *L* (m) is the membrane thickness, *m* (kg) is the mass of the molecule, *k* (m^2^∙kg∙s^−2^∙K^−1^) is Boltzmann’s constant, *T* (K) is the temperature, *I* (kg∙m^2^) is the moment of inertia, *N*_S_ (m^−3^) is the number of solubility sites, *N*_A_ (mol^−1^) is Avogadro’s number, *v** (s^−1^) is the vibrational frequency, and $\Delta E_{\mathrm{SS}}$ (J∙mol^−1^) is the activation energy. The jump distance, *d*, is related to *N*_S_ by
(9)$$d\;\left[nm\right]\;=\;\alpha Ns\;+\;\beta {\left(Ns\right)}^{2}\;+\;\gamma {\left(Ns\right)}^{3}\;+\;\delta {\left(Ns\right)}^{4}$$ where α = 0.84649, β = −1.74523 × 10^−29^ , γ = 5.60055 × 10^−58^, and δ = −7.66678 × 10^−87^, as reported in a previous study [40]. Importantly, the parameters in Equation (8) have physical significance to the permeating gas species. From permeance data, the jump distance, *d*, the number of solubility sites, *N*_S_, the vibrational frequency, *v**, and the activation energy, $\Delta E_{\mathrm{SS}}$, can be obtained, and it is later shown that these values fall within a range of realistic quantities.

The temperature dependence of permeance for each membrane was determined between 300 °C and 650 °C, which allowed for the gas permeance of several gas species to be fitted to the diffusion models discussed (Figure 10). The smallest gases—helium, neon and hydrogen—exhibited the best fit with the solid-state diffusion mechanism in all three membranes, which is consistent with previous results [40,43]. Additionally, the medium-sized molecules N_2_ and CO_2_ both showed a good fit with gas-translational diffusion. However, the larger-sized CH_4_ did not increase with temperature for the TMOS-derived membrane, suggesting that surface diffusion may be occurring.

Table 5 summarizes the hydrogen permeation parameters as an example of the He, Ne, and H_2_ permeances. Notably, the four parameters fit from the regression all fell within reasonable quantities. For instance, the number of solubility sites, *N_s_*, was in the range of 10^−26^ cm^−3^, which has an inverse cube root of ca. 10^−9^ cm. This would correspond to a jump distance, *d*, of ca. 1 nm, which is within the same order of magnitude as the values obtained, and suggests that these two parameters agree well. Furthermore the vibrational frequencies, *v**, were in the range of 10^12^ s^−1^, which is in the expected range for molecular vibrations. Lastly, the activation energies obtained and displayed in Table 5 were all less than 10 kJ∙mol^−1^, which is in the same range as that obtained for permeance of these gases in vitreous silica [44,45,46,47,48].

With regard to the changes between the three precursors, the jump distances and the number of solubility sites decreased with decreasing hydrogen permeance, but the effect on the number of solubility sites was clearly much more strongly tied to the total permeation value. Also, the vibrational frequencies in the TMOS- and MTMOS-derived membranes were similar, while that of the DMDMOS-derived membrane was a factor of about one-third smaller. The increasing activation energy can be correlated with two observations: FTIR measurements showed an increase in methyl groups, and SEM images indicated a higher activation energy for rougher surfaces. From this analysis, it appears that the addition of the methyl groups into the silica structure caused a wider and less dense pore structure, which resulted in an increase in the required energy for small molecules to permeate.

For the smaller gas molecules, CO_2_ and N_2_ both increased with temperature for all three membranes and fit well with the gas-translational diffusion mechanism (Table 6). CH_4_, however, showed an increase with temperature for the TMOS-derived membrane, which is more likely attributable to surface diffusion (Table 7). The DMDMOS-derived membrane appeared to have a low activation energy for CH_4_ permeation, suggesting it may have factors from both surface diffusion and gas-translational diffusion.

## 4. Conclusions

The objective of this research was to determine the effect of the number of methyl groups in a silica precursor on the synthesized membrane properties. Three siloxane compounds which had no, one, and two methyl groups were deposited. Before comparison of precursor differences, the residence time of each silica precursor in the CVD process was adjusted to improve the final gas permeance and perm-selectivity of the membrane. Increasing the number of methyl groups caused an increase in hydrothermal stability, along with an increase in the calculated activation energy for hydrogen permeation. FTIR spectroscopy verified the presence of methyl groups in the MTMOS- and DMDMOS-derived membranes, clearly showing a stronger presence in DMDMOS-derived than the MTMOS-derived membrane. These results indicate one of two possibilities: either the silica became denser with increasing methyl groups and caused a decrease in the accessible silicate species, or the methyl groups prevented the reconstruction of the silanol network.

## Figures and Tables

**Figure 1 membranes-09-00144-f001:**
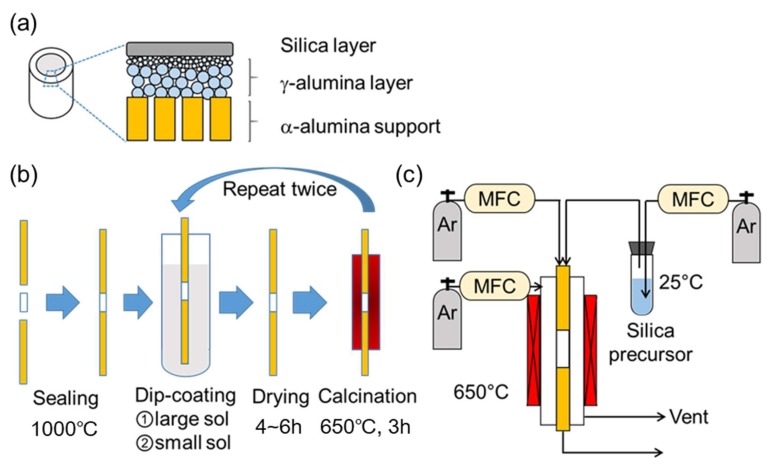
(**a**) Structure of the prepared silica membrane, (**b**) scheme of the dip-coating cycle, and (**c**) the chemical vapor deposition (CVD) apparatus.

**Figure 2 membranes-09-00144-f002:**
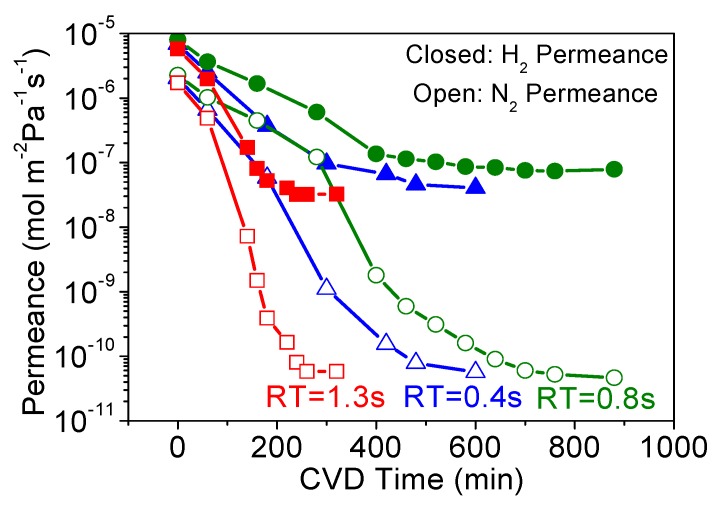
Changes in H_2_ (closed symbols) and N_2_ (open symbols) permeances during methyltrimethoxysilane (MTMOS) deposition at 650 °C with residence times of 0.4 s, 0.8 s, and 1.3 s.

**Figure 3 membranes-09-00144-f003:**
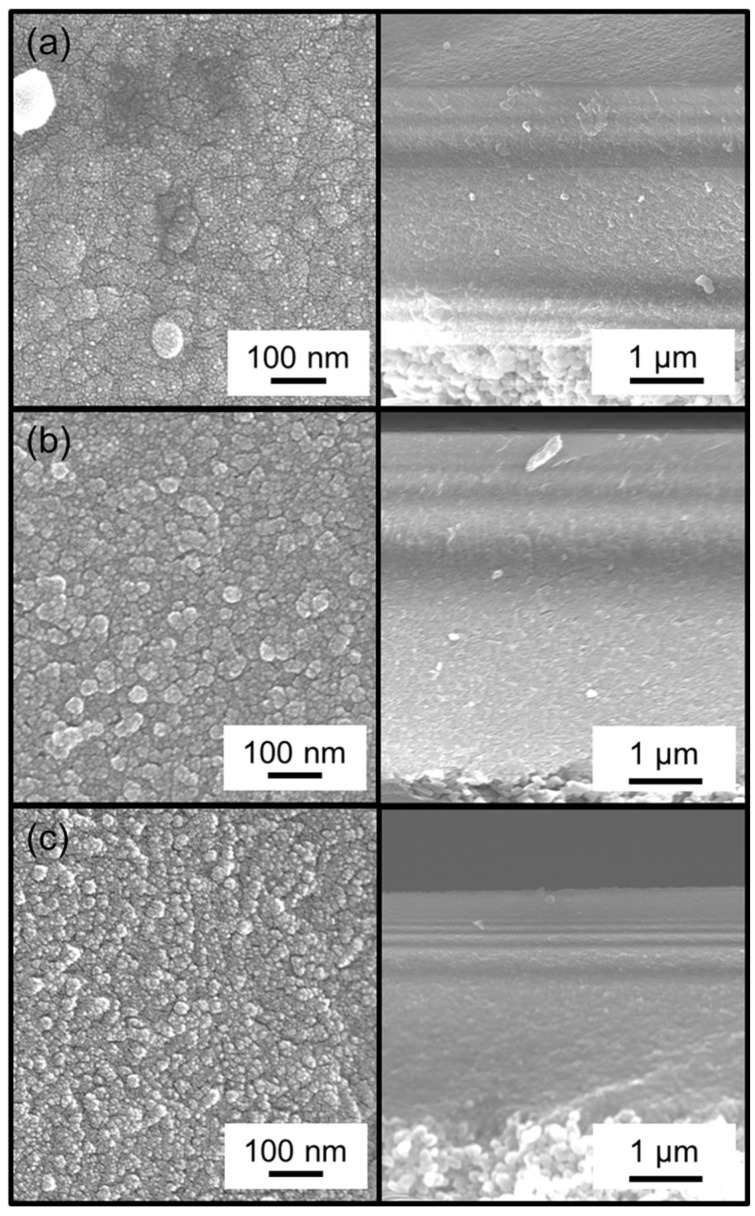
Surface (left) and cross-sectional (right) SEM images of the MTMOS-derived membranes deposited with residence times of (**a**) 0.4 s, (**b**) 0.8 s, and (**c**) 1.3 s.

**Figure 4 membranes-09-00144-f004:**
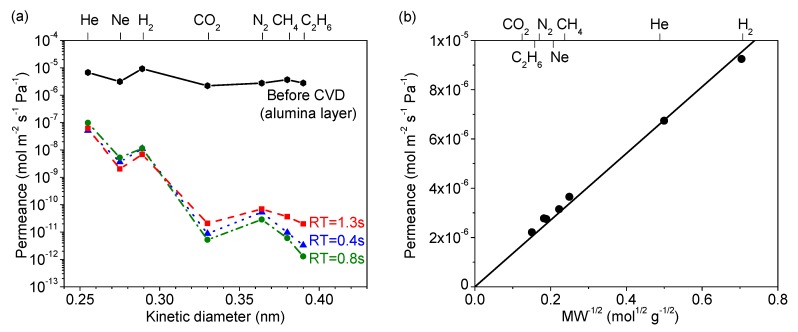
(**a**) Gas permeances at 300 °C of the MTMOS-derived membranes deposited at 650 °C with residence times of 0.4 s, 0.8 s, and 1.3 s. (**b**) Gas permeances at 300 °C through the alumina layer before CVD showing Knudsen selectivity.

**Figure 5 membranes-09-00144-f005:**
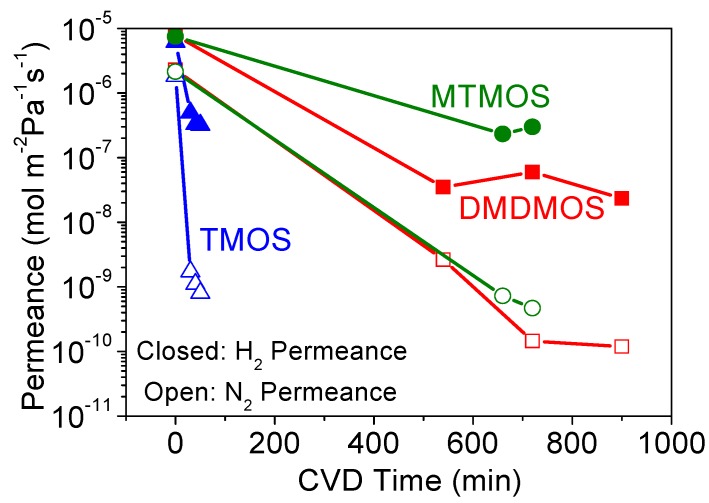
Changes in H_2_ (closed symbols) and N_2_ (open symbols) permeances during CVD at 650 °C of tetramethyl orthosilicate (TMOS), MTMOS, and dimethyldimethoxysilane (DMDMOS) precursors.

**Figure 6 membranes-09-00144-f006:**
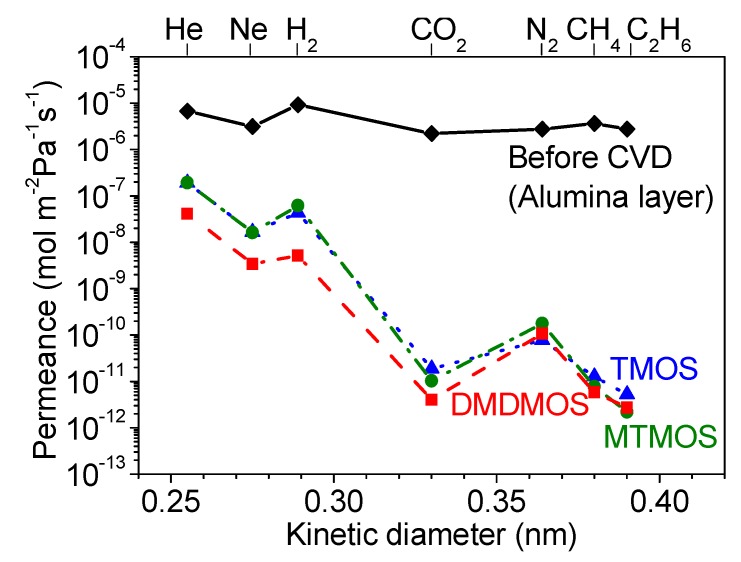
Gas permeances at 300 °C through the (◆) alumina intermediate layer, (▲) TMOS-, (●) MTMOS-, and (■) DMDMOS-derived membranes.

**Figure 7 membranes-09-00144-f007:**
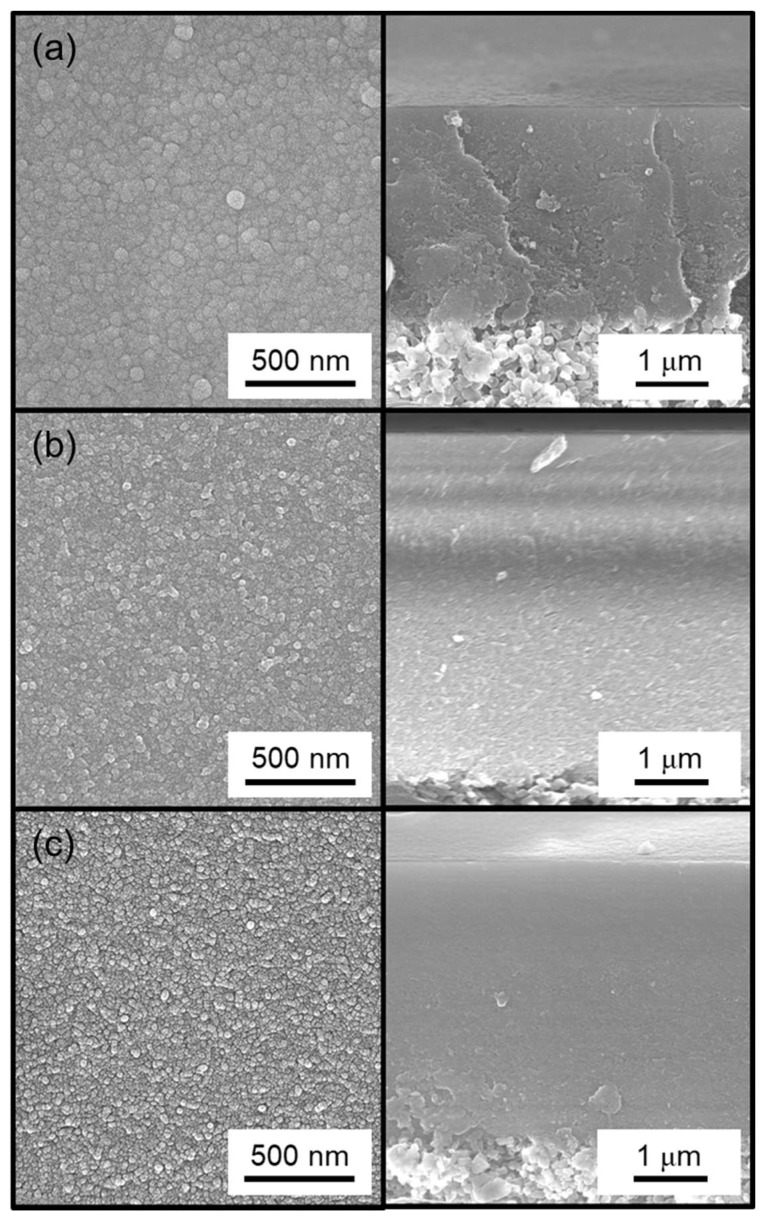
Surface (left) and cross-sectional (right) SEM images of the (**a**) TMOS-derived membrane, (**b**) MTMOS-derived membrane, and (**c**) DMDMOS-derived membrane.

**Figure 8 membranes-09-00144-f008:**
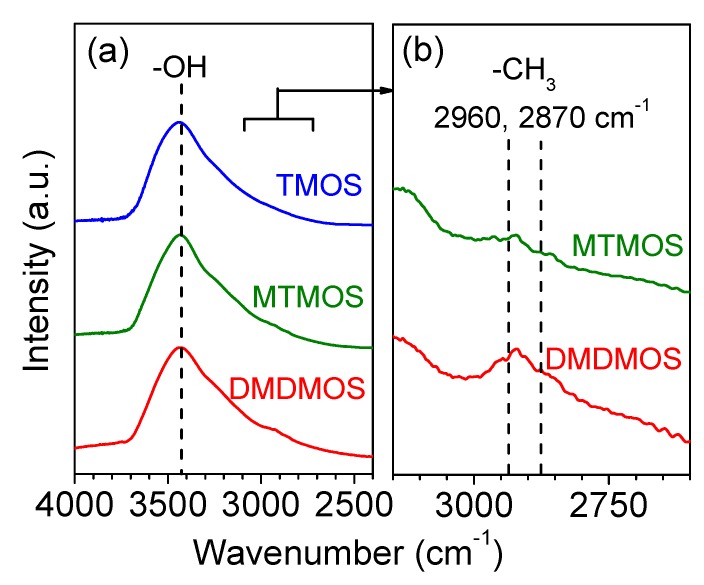
(**a**) Fourier-transform infrared (FTIR) spectra of the TMOS-, MTMOS-, and DMDMOS-derived membranes, and (**b**) the spectra of MTMOS- and DMDMOS-derived membranes after subtracting the spectra of the TMOS-derived membrane.

**Figure 9 membranes-09-00144-f009:**
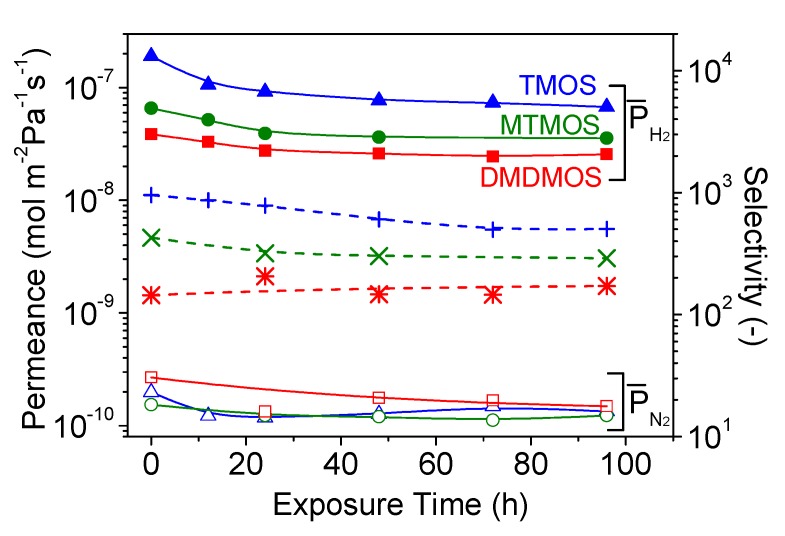
Changes in H_2_ and N_2_ permeances and selectivity through the TMOS-, MTMOS-, and DMDMOS-derived membranes under 15 mol.% H_2_O at 650 °C.

**Figure 10 membranes-09-00144-f010:**
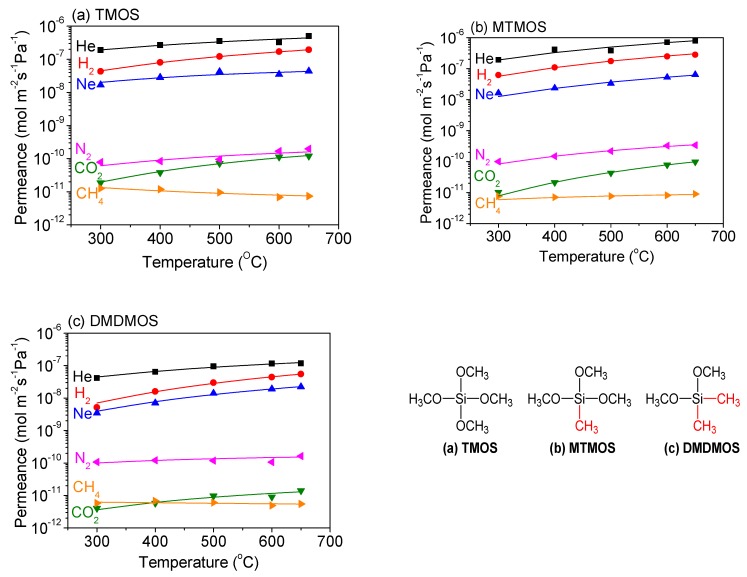
Temperature dependence of the permeance for several gas species through (**a**) TMOS-, (**b**) MTMOS-, and (**c**) DMDMOS-derived membranes. Lines represent the fitted results.

**Table 1 membranes-09-00144-t001:** Silica precursors and their vapor pressures.

Silica Precursor	Tetramethyl Orthosilicate (TMOS)	Methyltrimethoxysilane (MTMOS)	Dimethyldimethoxysilane (DMDMOS)
Chemical structure	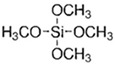	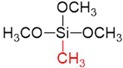	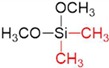
Vapor pressure (atm) (25 °C)	0.023	0.051	0.12

**Table 2 membranes-09-00144-t002:** Summary of the membrane performance at 300 °C, and deposition conditions of the MTMOS-derived membranes deposited with residence times of 0.4 s, 0.8 s, and 1.3 s.

Membrane	H_2_ Permeance	N_2_ Permeance	H_2_/N_2_ Selectivity	CVD Time	Precursor Flow Rate	Total Precursor Flowed
	(mol∙m^−2^∙s^−1^∙Pa^−1^)		(–)	(min)	(μmol∙s^−1^)	(mmol)
RT = 0.4 s	1.12 × 10^−8^	5.3 × 10^−11^	210	600	0.46	17
RT = 0.8 s	1.14 × 10^−8^	2.9 × 10^−11^	400	880	0.23	12
RT = 1.3 s	6.8 × 10^−9^	6.9 × 10^−11^	99	320	0.15	2.9

**Table 3 membranes-09-00144-t003:** Comparison of data from Nomura et al. [33] with the current work.

Precursor	Study	H_2_ Permeance (mol∙m^−2^∙s^−1^∙Pa^−1^)(600 °C)	H_2_/N_2_ Selectivity (–) (600 °C)
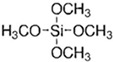 TMOS	Nomura et al.	2 × 10^−7^	610
	This work	1.7 × 10^−7^	990
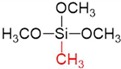 MTMOS	Nomura et al.	3 × 10^−7^	590
	This work	2.4 × 10^−7^	740
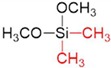 DMDMOS	Nomura et al.	9 × 10^−7^	920
	This work	4.4 × 10^−8^	410

**Table 4 membranes-09-00144-t004:** Summary of the percentage change of the TMOS-, MTMOS-, and DMDMOS-derived membranes after 96 h exposure to 16 mol.% H_2_O at 650 °C.

	TMOS	MTMOS	DMDMOS
H_2_ Permeance	−64%	−46%	−34%
N_2_ Permeance	−32%	−20%	−44%
H_2_/N_2_ Selectivity	−47%	−32%	+19%

**Table 5 membranes-09-00144-t005:** Solid-state diffusion model Equation (8) factors for hydrogen permeation through the TMOS-, MTMOS-, and DMDMOS-derived membranes.

	TMOS	MTMOS	DMDMOS
Jumping distance, *d* (nm)	0.75	0.79	0.72
Number of solubility sites, *N*_S_ (10^25^ m^−3^)	7.3	9.7	0.75
Vibration frequency, *v* (10^12^ s^−1^)	1.9	1.8	0.68
Activation energy, Δ*E* (kJ∙mol^−1^)	2.0	2.1	2.9
Regression coefficient, *R*^2^ (–)	0.999	0.999	0.995

**Table 6 membranes-09-00144-t006:** Model factors calculated by fitting the membrane permeances to the gas-translational diffusion model Equation (7).

Silica Precursor	Gas	Constant, *C* (mol∙m^−2^∙s^−1^∙Pa^−1^∙K^−1/2^)	Activation Energy, ∆*E*_a_ (kJ∙mol^−1^)	Regression Coefficient, *R*^2^ (–)
TMOS	CO_2_	1.2 × 10^−7^	26	0.997
	N_2_	3.5 × 10^−8^	15	0.955
MTMOS	CO_2_	2.8 × 10^−7^	35	0.981
	N_2_	1.6 × 10^−7^	21	0.998
	CH_4_	7.4 × 10^−10^	7.9	0.999
DMDMOS	CO_2_	4.7 × 10^−9^	19	0.975
	N_2_	1.4 × 10^−8^	8.3	0.991
	CH_4_	2.0 × 10^−10^	1.3	0.991

**Table 7 membranes-09-00144-t007:** Model factors calculated by fitting the permeances through the TMOS-, MTMOS-, and DMDMOS-derived membranes to the surface diffusion model Equation (6).

Silica Precursor	Gas	Kinetic Diameter (nm)	${\bar{P}}_{0}$ (mol∙m^−2^∙s^−1^∙Pa^−1^)	−∆*H*_a_ −∆*E*_a_ (kJ∙mol^−1^)	Regression Coefficient, *R*^2^ (–)	Enthalpy of Vaporization, ∆*H*_vap_ (kJ∙mol^−1^)
TMOS	CH_4_	0.38	2.9 × 10^−12^	7.3	0.990	8.2
DMDMOS	CH_4_	0.38	4.4 × 10^−12^	1.7	0.990	8.2

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
