# Peer review of "Gas Separation Silica Membranes Prepared by Chemical Vapor Deposition of Methyl-Substituted Silanes"

_membranes, 2019, doi:10.3390/membranes9110144_

Round 1

Reviewer 1 Report

In this manuscript the authors used three different silanes as reactants for CVD to fabricate thin silica membranes for gas separation. The gas permeation performance of these membranes was evaluated and discussed. I have several questions and suggestions as below:  

Line 55 to 57, 3 nm and 0.49 nm should switch their places. In the introduction section you mentioned the work by Nomura et al. who also studied different silica precursors including the ones you have used in this work. Could you please describe more about the difference and novelty of your work compared to Nomura’s work? Is there any reference for equation (3)? Typos in line 206 and 213. Line 221 to line 223, “From these results, the residence time for the DMDMOS- and TMOS-depositions were chosen to be 0.6 s for DMDMOS and 0.8 s for TMOS.” By saying this, do you mean you have used the H2/N2 selectivity as criteria to determine the optimum resistance time for DMDMOS and TMOS also? How did you determine the membrane thickness of 50 nm? It is not easy to identify the membrane layer from the SEM images. Could you also show the data of TMOS in Figure 8 (b)? The H2 permeance from Nomura’s work (9×10-7 mol m-2 s-1 Pa-1, selectivity 920) is much higher than that from your work (2.4×10-7 mol m-2 s-1 Pa-1, selectivity 740). What do you think is critical to the different performance?

Reviewer 2 Report

Comments

Page 2, Line 56: "the estimated pore size in the membrane, increasing from ca. 3 nm for the tetraethyl orthosilicate-56 derived membrane to ca. 0.49 nm for...". Here, it seems pore size is decreasing from 3 to 0.49 nm rather than increasing. In this case, the authors should clarify the explanation given in the manuscript based on the increase in pore size with increase in phenyl groups, which is not the case.

Page 4, Line 56: Equation (3) should be revisited. As mentioned in the manuscript, “?̅?????? ???, was used as the permeation resistance of the support.” Thus, P after CVD is the permeance resistance of the support plus silica layer. In this case, the latter must be greater than the  ?̅?????? ???In  this eq (3) should be the following:

If the above argument is valid, the authors should clarify the correctness of the results provided in the manuscript. Also, I suggest the authors to provide the reference to permeation resistance.

Page 5, Line 180: “Although the final N2 permeance did not vary significantly, the required CVD time was significantly different for each retention time and there was no trend in the results, suggesting a tradeoff between different factors….”!! A more detailed explanation highlighting the factors that plays a role in this case is beneficial to the less informed user.

Page 7, Line 206: Check the units “all the gases were very high (10-5 to 10-6 mol m+s+Pa-1).”

Page 7, Line 213: Typo:  change themembrane to the membrane

Page 7, Line 217: “RT=0.8 s in H2 permeance (3.2×10-7 vs. 3.0×10-7 mol m-2 s-1 Pa-1) and H2/N2 selectivity (400 vs. 640) ..”  From the graph it seems, the permeance of H2  is same at all RTs, while  the permeance of N2  is same when RT=0.4 s and RT =1.3s.  In this case the authors should clarify why the H2/N2 selectivity (400 vs. 640) is very different when RT=0.4 s and 0.8 s. Further, the authors should include the error bars in their calculations. They also need to provide how many independent experiments were conducted at an any given point?

Table -2: The membrane thickness column should be removed as they were not sure about that.

Page 8, Line 234: Typo - Figure 5 and 6. Figure 6.

Table 3: Authors must provide a detailed explanation for the contradiction between the results of present work and the literature data, as presented in Table -3. For example what synthesis conditions or membrane thickness with proper references.

Reviewer 3 Report

The manuscript presents a study on the gas transport through silica membranes prepared by CVD using silica precursors with different number of methyl groups. The effect of the 50 nm thick film on the test gas transport properties was examined by gas permeation method.

The topic is of interest and the research seems me well conducted.

I think anyway that the discussion on the obtained permeance data has to be graetly improved. I have many questions about the gas transport properties. Some of them:

Can the authors explain me how eq. 3 is obtained? Which are the physical assumptions behind it? In particular, how does the model consider the mass transfer processes at the support-CVD layer interface? The authors claim that the transport of small molecules (H2, He and Ne) through the CVD layer is controlled by a solid state diffusion model (eq. 8 which contains a refuse, I think). Can they tell me what are the solution sites for those molecules? Moreover I would like them comment on the implications of this transport process on the physical state of the soluted H2 molecule. Does this molecule dissociate or not in its solution process? Eq. 8 is quite complicated. Which are the free parameters in it for data fitting? Why do the authors present and analyze permeance values instead of permeability ones? Why do the authors present permeance values vs temperature and not their Arrhenius plot? I would like that the authors comment on the numerical values obtained by permeance fitting. This is important to understand if the obtained activation energy values, frequency factors, .... are compatible with the presented model.

Round 2

Reviewer 2 Report

The authors have addressed all of my concerns except the following:

Comment#2:

Authors reply

We have modified the wording to clarify that ?̅????????? is still a permeance, and its inverse is the quantity used as the resistance value. Because it is the permeance before CVD, it has a higher value than ?̅After???

Comment: The authors mentioned that  the permeance before CVD ( ?̅????????? ) has a  higher value than  the permeance after CVD  ?̅After???

????????? > ?̅After???

which means 

(1/?̅?????????) <  (1/?̅After???)

Then, in their eq (3), 

1/?̅Silica Layer =(1/?̅?????????)- (1/?̅After???)

the first term is lower than second term, leading to the resistance offered by silica layer is negative, which seems wrong to me. This needs to be addressed.

Author Response

Thank you for your comments. The Eq. 3 has been corrected in the revised version.